# Interplay of hidden orbital order and superconductivity in CeCoIn$_5$

Weijiong Chen [1,11], Clara Neerup Breiø[2,11], Freek Massee[3,11], Milan P. Allan [4], Cedomir Petrovic [5], J. C. Séamus Davis [1,6,7,8] ✉, Peter J. Hirschfeld[9], Brian M. Andersen[2] ✉ & Andreas Kreisel [2,10]

Visualizing atomic-orbital degrees of freedom is a frontier challenge in scanned microscopy. Some types of orbital order are virtually imperceptible to normal scattering techniques because they do not reduce the overall crystal lattice symmetry. A good example is $d_{xz}/d_{yz}$ (π,π) orbital order in tetragonal lattices. For enhanced detectability, here we consider the quasiparticle scattering interference (QPI) signature of such (π,π) orbital order in both normal and superconducting phases. The theory reveals that sublattice-specific QPI signatures generated by the orbital order should emerge strongly in the superconducting phase. Sublattice-resolved QPI visualization in superconducting CeCoIn$_5$ then reveals two orthogonal QPI patterns at lattice-substitutional impurity atoms. We analyze the energy dependence of these two orthogonal QPI patterns and find the intensity peaked near $E = 0$, as predicted when such (π,π) orbital order is intertwined with $d$-wave superconductivity. Sublattice-resolved superconductive QPI techniques thus represent a new approach for study of hidden orbital order.

In a crystalline metal, strong electronic correlations occurring between electrons derived from different orbitals in the same atom can yield an orbital-selective Hund's metal state[1,2], or even orbital-selective superconductivity[3–5]. Similarly, symmetry breaking orbital order may occur, with one of the most famous cases being in the Fe-based high-temperature superconductors[6,7]. However, some types of orbital order are almost indiscernible because they do not occur with any lattice distortion, which reduces the overall crystal lattice symmetry. For example, (π,π) orbital order in a tetragonal array of transition-metal atoms occurs when the degeneracy of $d_{xz}$ and $d_{yz}$ orbitals is lifted and each predominates energetically over the other at alternating lattice sites (Fig. 1a). This subtle state does not alter the crystal lattice symmetry meaning that it is virtually invisible to

normal photon and neutron scattering techniques, since these techniques are mainly sensitive to the core electron scattering and the nuclear scattering, respectively[8,9]. By contrast, conventional scanning tunneling microscopy (STM) has reported evidence for (π,π) orbital order on the surface of CeCoIn$_5$[10], revealing an opportunity for quasiparticle scattering interference (QPI) imaging, a powerful technique for detecting subtle orbital-selective effects[3,11].

The QPI effect[12,13] occurs when an impurity atom/vacancy scatters quasiparticles, which then interfere to produce characteristic modulations of the density-of-states $\delta N(\mathbf{r},E)$ surrounding each impurity site. Impurity scattering is usually studied by using $|\delta N(\mathbf{q},E)|$, the square root of the power-spectral-density Fourier transform of the

[1]Clarendon Laboratory, University of Oxford, Oxford OX1 3PU, UK. [2]Niels Bohr Institute, University of Copenhagen, 2100 Copenhagen, Denmark. [3]Laboratoire de Physique des Solides (CNRS UMR 8502), Bâtiment 510, Université Paris-Sud/Université Paris-Saclay, 91405 Orsay, France. [4]Leiden Institute of Physics, Leiden University, P.O. Box 9504, 2300 RA Leiden, The Netherlands. [5]CMPMS Department, Brookhaven National Laboratory, Upton, NY 11973, USA. [6]LASSP, Department of Physics, Cornell University, Ithaca, NY 14850, USA. [7]Department of Physics, University College Cork, Cork T12 R5C, Ireland. [8]Max-Planck Institute for Chemical Physics of Solids, 01187 Dresden, Germany. [9]Department of Physics, University of Florida, Gainesville, FL 32611, USA. [10]Inst. für Theoretische Physik, Universität Leipzig, Brüderstr. 16, Leipzig 04103, Germany. [11]These authors contributed equally: Weijiong Chen, Clara Neerup Breiø, Freek Massee. ✉e-mail: jcseamusdavis@gmail.com; bma@nbi.ku.dk

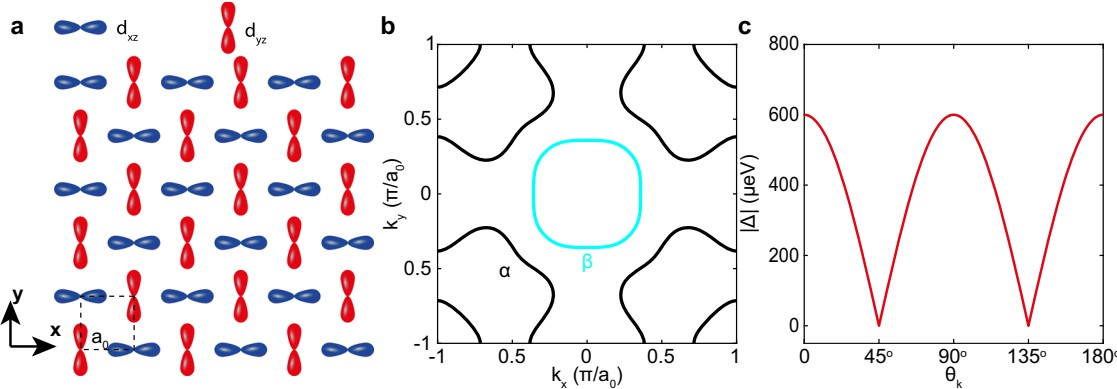

**Fig. 1 | $(\pi, \pi)$ orbital order on the surface of CeCoIn$_5$. a** Schematic of $(\pi, \pi)$ orbital order on the surface of CeCoIn$_5$. Two sublattices are introduced by the $d_{xz}/d_{yz}$ orbital order. **b** The Fermi surface of CeCoIn$_5$ measured by heavy-fermion quasiparticle interference[19]. **c** Superconducting energy gap structure of CeCoIn$_5$ measured about the $(\pi, \pi)$ point[19]. The order parameter is believed to exhibit $d_{x^2-y^2}$ symmetry.

perturbation to the density of states by the impurity

$$\delta N(\mathbf{q}, E) = -\frac{1}{\pi} \mathrm{Tr}\left(\mathrm{Im}\sum_{\mathbf{k}}(G(\mathbf{k}, E+i\eta)T(E)G(\mathbf{k}+\mathbf{q}, E+i\eta))\right) \quad (1)$$

Here, $G(\mathbf{k}, E+i\eta)$ is the electron propagator $G(\mathbf{k}, E+i\eta) = 1/(E+i\eta - E_0(\mathbf{k}) - \Sigma(\mathbf{k}, E+i\eta))$ of a quasiparticle state $|\mathbf{k}\rangle$ with momentum $\mathbf{k}$, and $\Sigma(\mathbf{k}, E+i\eta) = \mathrm{Re}\Sigma(\mathbf{k}, E+i\eta) + i\mathrm{Im}\Sigma(\mathbf{k}, E+i\eta)$ is the self-energy of interacting electrons. $T(E)$ is the so-called T-matrix, representing the possible scattering processes between states $|\mathbf{k}\rangle$ and $|\mathbf{k}+\mathbf{q}\rangle$ for a point-like s-wave scatterer. Atomic scale imaging of these interference patterns $\delta N(\mathbf{r}, E)$ is achieved using spatial mapping of the differential conductance, $g(\mathbf{r}, E)$[14].

## Results

### Modeling the $(\pi, \pi)$ orbital order

As a concrete model, we consider orbital order of $d_{xz}/d_{yz}$-orbitals on a 2D square-lattice (Fig. 1a). To accommodate the $(\pi, \pi)$ orbital order, the unit cell is enlarged to a two-sublattice basis allowing for the incorporation of a staggered, nematic orbital order preserving the translational and global $C_4$-symmetry. Including superconductivity the model Hamiltonian takes the form,

$$H = \sum_{\mathbf{k}} \psi_{\mathbf{k}}^{\dagger} \begin{pmatrix} \mathcal{H}_0(\mathbf{k}) + \mathcal{H}_{oo}(\mathbf{k}) & \Delta_d(\mathbf{k}) \\ \Delta_d^{\dagger}(\mathbf{k}) & -\mathcal{H}_0^*(-\mathbf{k}) - \mathcal{H}_{oo}^*(-\mathbf{k}) \end{pmatrix} \psi_{\mathbf{k}}, \quad (2)$$

where the Nambu spinor is defined as

$$\psi_{\mathbf{k}} = \left(c_{A,xz,\uparrow}(\mathbf{k}), c_{A,yz,\uparrow}(\mathbf{k}), c_{B,xz,\uparrow}(\mathbf{k}), c_{B,yz,\uparrow}(\mathbf{k}), c_{A,xz,\downarrow}^{\dagger}(-\mathbf{k}), c_{A,yz,\downarrow}^{\dagger}(-\mathbf{k}), c_{B,xz,\downarrow}^{\dagger}(-\mathbf{k}), c_{B,yz,\downarrow}^{\dagger}(-\mathbf{k})\right)^T$$

with $c_{\nu,\mu,\sigma}(\mathbf{k})$ annihilating an electron with momentum $\mathbf{k}$ and spin $\sigma$ at sublattice $\nu$ in orbital $d_\mu$. Here $\mathcal{H}_0(\mathbf{k})$ contains intra- and interorbital nearest- and next-nearest-neighbor hoppings allowed by the d-wave symmetry of the orbitals, $\mathcal{H}_{oo}(\mathbf{k})$ introduces the on-site anti-ferro-orbital order and $\Delta_d(\mathbf{k})$ contains nearest- and next-nearest-neighbor intraorbital d-wave pairings as introduced in ref. 15. Here for generality we consider the simplest model Hamiltonian ($\mathcal{H}_0(\mathbf{k}), \mathcal{H}_{oo}(\mathbf{k})$) rather than specific Hamiltonian of CeCoIn$_5$. As the model is spin-independent we suppress the spin index below. To separate the energy scales of the orbital and superconducting orders, the orbital order is introduced at an energy scale well above the superconducting gap, i.e., $\Delta_{oo} \gg \Delta_d$. The Hamiltonian in (2) is chosen as a minimal model approach where $\mathcal{H}_0(\mathbf{k})$ describes the simplest band dispersion allowing for the implementation of local $C_4$-symmetry breaking. A detailed description of the model and parameters can be found in SI Section 1.

To simulate QPI, a non-magnetic impurity is introduced as a point-like potential. We choose an on-site impurity as the scattering center, because this kind of impurity widely exists in the crystals and is located at a high-symmetry point required to detect the local symmetry breaking caused by the orbital order. The impurity, either a different element or lattice vacancy, is assumed to exhibit a trivial spatial structure leading to identical potential strengths in the orbital degree of freedom. The local density of states (LDOS) is computed using a T-matrix approach as

$$N(\mathbf{R}, \gamma, E) = -\frac{1}{\pi} \mathrm{Im}\left(G^R(\mathbf{0}, E) + G^R(\mathbf{R}, E)T(\mathbf{0}, E)G^R(-\mathbf{R}, E)\right)_{\gamma\gamma} \quad (3)$$

where $\mathbf{R}$ is the real-space position of the two-ion unit cell, $\gamma \in \{\nu = A, B; \mu = xz, yz\}$, the T-matrix is given by $T(\mathbf{0}, E) = [1 - H_{imp}(\mathbf{0})G^R(\mathbf{0}, E)]^{-1}H_{imp}(\mathbf{0})$ and $G^R(\mathbf{R}, E) = \mathcal{G}^0(\mathbf{R}, i\omega_n \to E + i\eta) = \sum_{\mathbf{k}} e^{i\mathbf{k}\cdot\mathbf{R}}\mathcal{G}^0(\mathbf{k}, i\omega_n)$ is the bare, retarded Greens function obtained from (2). We always insert the impurity at one of the two sites in the unit cell positioned at $\mathbf{R} = \mathbf{0}$ for simplicity. Note that $N(\mathbf{R}, \gamma, E)$ contains four components for the unit cell at $\mathbf{R}$, corresponding to the orbital and sublattice degrees of freedom. The position of a single lattice site, $\mathbf{r}$, is uniquely mapped from the set $\{\mathbf{R}, \nu\}$ enabling a straightforward transition to the site-resolved LDOS. To allow for reliable comparison to experimental data, we calculate the local density of states above the surface of the material following a simplified method that takes the Wannier orbitals into account[16,17] and basically weigh the computed $N(\mathbf{r}, E)$ by atomic-like $d_{xz}/d_{yz}$ orbitals. To account for the experimental resolution of 100 µeV, we perform an additional Gaussian energy convolution, details on these calculations can be found in SI Section 2.

### Consequences for QPI of $(\pi, \pi)$ orbital order

The consequences of this $(\pi, \pi)$ orbital order for QPI experiments are intriguing. Surprisingly, theoretical modeling for the $\mathbf{r}$-space QPI patterns, $N(\mathbf{r}, E)$, around the impurity at sublattice *a* (Fig. 2a) and sublattice *b* (Fig. 2d) at energy $|E| > \Delta$ well outside the superconducting gap, show almost identical $N(\mathbf{r}, E)$. At energies $|E| < \Delta$, however, the situation is radically different. Here $N(\mathbf{r}, E)$ around chemically identical impurity atoms at sublattice *a* (Fig. 2b) and sublattice *b* (Fig. 2e) are vividly different. The key consequence is that the amplitude of scattering interference is far more intense along one axis than along the other axis, depending on which sublattice the impurity atom resides. The interference pattern breaks $C_4$-rotational symmetry, indicating the existence of the hidden $(\pi, \pi)$ orbital order, which breaks $C_4$-symmetry locally. We stress that the impurity potential itself is point-like and of identical strength on both orbitals. The $C_4$-symmetry breaking takes

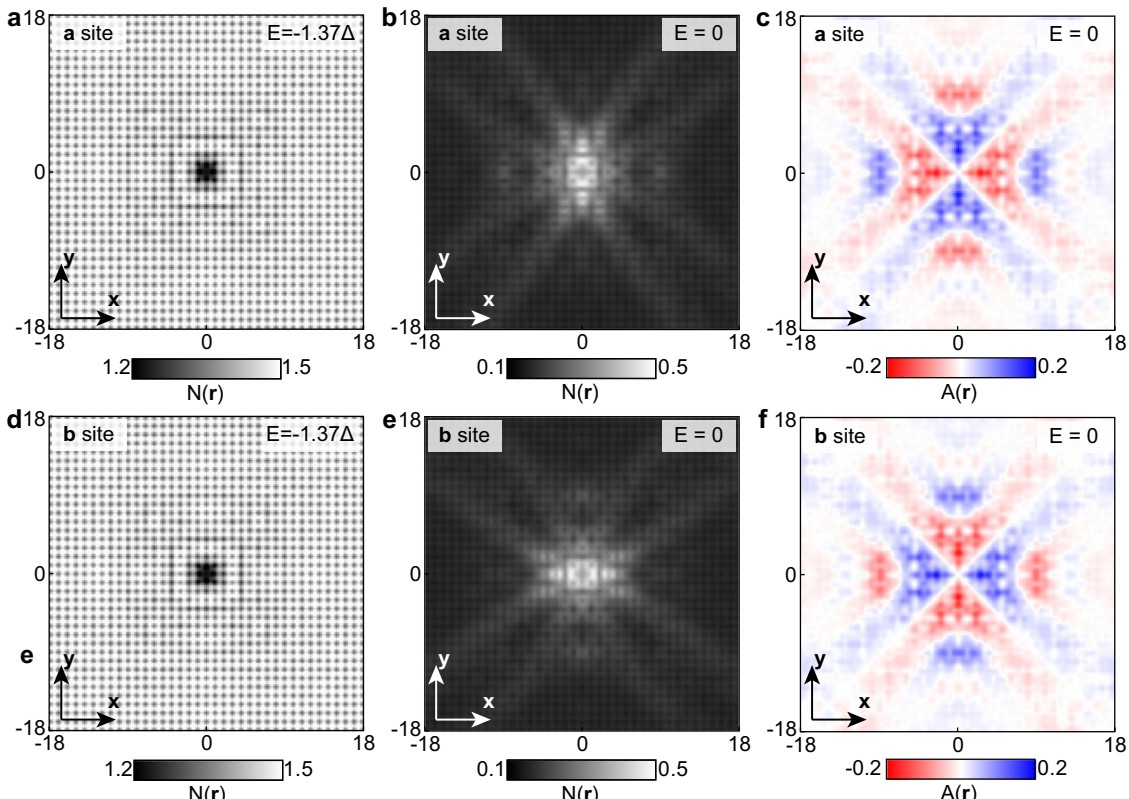

**Fig. 2 | Bogoliubov quasiparticle interference from (π, π) orbital order calculated by the theoretical models. a, d** Theoretical results for BQPI pattern $N(\mathbf{r},E)$ with the impurity atom at sublattice $\boldsymbol{a}$ (**a**) and sublattice $\boldsymbol{b}$ (**d**) at the energy well outside the superconducting gap $E>|\Delta|$. **b, e** Theoretical results for BQPI pattern $N(\mathbf{r},E)$ with the impurity atom at sublattice $\boldsymbol{a}$ (**b**) and sublattice $\boldsymbol{b}$ (**e**) at the energy well below the superconducting gap edge $E<|\Delta|$. **c, f** The local anisotropy $A(\mathbf{r},E)$ with the impurity atom at sublattice $\boldsymbol{a}$ (**c**) and sublattice $\boldsymbol{b}$ (**f**) at the energy well below the superconducting gap edge $E<|\Delta|$.

place because the impurity chooses a specific sublattice, in conjunction with the underlying orbital order. To quantify this local symmetry breaking effect, we define a dimensionless value $A(\mathbf{r},E) = (N(\mathbf{r},E) - N^{\circlearrowleft 90}(\mathbf{r},E))/(N(\mathbf{r},E)+N^{\circlearrowleft 90}(\mathbf{r},E))$ as the local anisotropy, in which $N^{\circlearrowleft 90}(\mathbf{r},E)$ is 90-degree anti-clockwise-rotated $N(\mathbf{r},E)$ surrounding the impurity site at sublattice $\boldsymbol{a}/\boldsymbol{b}$. The $A(\mathbf{r},E)$ maps (Fig. 2c, f) at energies $|E|<\Delta$ explicitly demonstrate the $C_4$-symmetry breaking for both impurity positions. The maximum $A(\mathbf{r},E)$ value approaches 20%. Meanwhile, at the energy $|E|>\Delta$, still within the energy scale of the orbital order ($\Delta_{oo}$), $A(\mathbf{r},E)$ is less than 2% (Fig. S2). Thus, the orbital order can be clearly unraveled below the energy scale of the superconducting order parameter because the opening of the superconducting gap selectively enhances its visibility. For comparison, the QPI simulation of the normal-state model at the energies $|E|<\Delta$ can be seen Fig. S3, which is equivalent to the $|E|>\Delta$ case of the superconducting model.

## QPI signature of (π,π) orbital order in CeCoIn$_5$

To explore these predictions we studied CeCoIn$_5$, a prototypical heavy-fermion superconductor, whose crystal unit cell has dimensions $a = b = 4.6$ Å, $c = 7.51$ Å and with superconducting critical temperature $T_c = 2.3$ K (ref. 18). As revealed by heavy-fermion scattering interference imaging, its Fermi surface is formed by two heavy bands ($\alpha$ and $\beta$ bands in Fig. 1b) due to the hybridization of a conventional light conduction band and the localized f-electrons[19]. In the superconducting state, the Cooper pairs are spin-singlets[20,21] and a Cooper pairing energy gap with apparent nodes $|\Delta_{\alpha}(\mathbf{k})| = 0$ oriented along the $\mathbf{k} = [(1,1);(1,-1)]2\pi/a$ directions[21–25] and a nodal, V-shaped $N(E)\propto E$ with gap edges $600 \pm 50$ μeV. The $|\Delta_{\alpha}(\mathbf{k})|$ measured in **k**-space with QPI is shown in Fig. 1c[19]. Our CeCoIn$_5$ single crystal samples are inserted into the

spectroscopic imaging STM, cleaved in cryogenic ultra-high vacuum, inserted into the STM head and cooled to $T = 280$ mK.

A standard Co terminated surface topography $T(\mathbf{r})$ is shown in Fig. 3a with sublattices marked by red dots and blue dots, respectively. The Co terminated surface is identified from both the tunneling conductance spectrum and the domain boundaries (SI section 4). In this field of view (FOV), we find two single atom defects allocated at sublattice $\boldsymbol{a}$ and $\boldsymbol{b}$, respectively. These two defects are nearly identical in topography image (Fig. 3a). Figure 3b shows simultaneously measured differential conductance map $g(\mathbf{r},E)$ at $E = -0.94$ meV ($E>|\Delta|$). Virtually, no difference in scattering interferences from defects in the different sublattices can be detected. In contrast, the simultaneously measured differential conductance map $g(\mathbf{r},E)$ at $E = 0$ in the same FOV shown in Fig. 3c reveals highly distinct interference patterns. The scattering interference of one defect is far more intense along the $\boldsymbol{a}$ axis than the $\boldsymbol{b}$ axis, and vice versa. Indeed, they appear to be rotated by 90-degrees relative to each other, in agreement with the theoretical prediction in Fig. 2. Furthermore, Fig. S5 gives the comparison of $A(\mathbf{r},E)$ surrounding the same defect in the superconducting state ($T<T_c$) (Fig. S5a–c) and in the normal state ($T>T_c$) (Fig. S5d–f). The local anisotropy $A(\mathbf{r},E)$ is only enhanced at E = 0 in the superconducting state while has no apparent change at $E = 0$ in the normal state, in agreement with the theoretical prediction in Fig. S4.

Next, we study the local anisotropy $A(\mathbf{r},E)$ around the defects at the two sublattices. Figure 4a,d contain the measured local anisotropy $A(\mathbf{r},E)$ at $E = 0$ around the defects at sublattice $\boldsymbol{a}$ (Fig. 4a) and sublattice $\boldsymbol{b}$ (Fig. 4d). Obviously, the conductance anisotropy is rotated by 90-degrees for scattering centers at the different sublattice sites. To analyze the energy-dependence of $A(\mathbf{r},E)$ we plot in Fig. 4b, e, the line profiles of $A(\mathbf{r},E)$ along the high-symmetry directions (0,1) and (1,0)

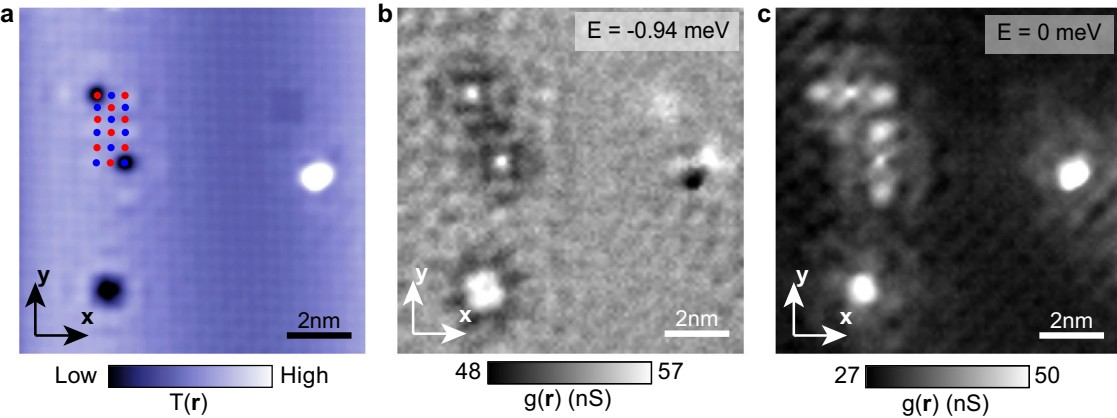

**Fig. 3 | Example of QPI imaging resolved sublattices in CeCoIn₅. a** Atomic resolved topography image around two sublattices. Two sublattices are indicated schematically by red dots and blue dots, respectively. (setpoint: $V = -10meV, I = 800pA$). **b** Simultaneous measured differential conductance map

$g(\mathbf{r}, E)$ at $E = -0.94meV$ in the FOV of image (**a**). **c** Simultaneous measured differential conductance map $g(\mathbf{r}, E)$ at $E = 0$ in the FOV of image (**a**). The BQPI patterns on the two sublattices are clearly distinct and appear to be rotated by 90-degrees relative to each other.

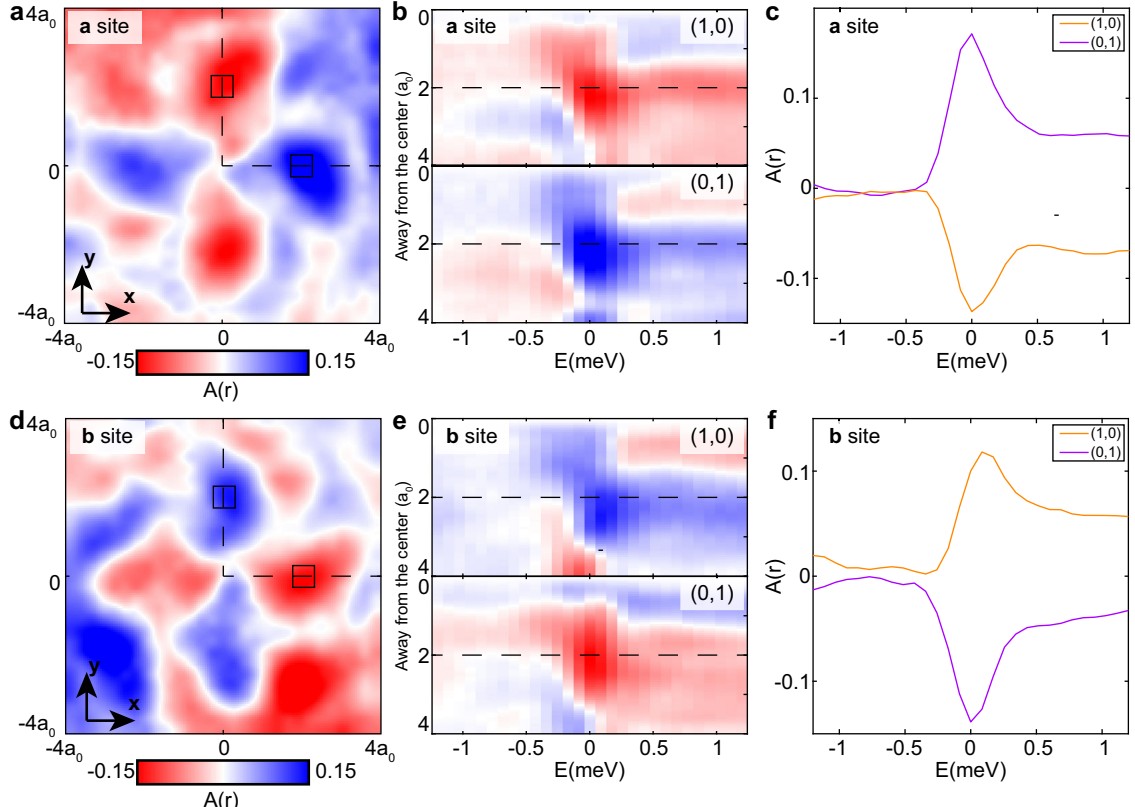

**Fig. 4 | Local anisotropy $A(\mathbf{r}, E)$ around defects in two sublattices. a, d** Measured local anisotropy $A(\mathbf{r}, E)$ around the defects at sublattice $a$ (**a**) and sublattice $b$ (**d**). The length scale of **a, d** is in the unit of lattice constant $a_0$. **b, e** Measured local anisotropy $A(\mathbf{r}, E)$ around the defects at sublattice $a$ (**b**) and sublattice $b$ (**e**) along

the high-symmetry direction (1,0) and (0,1) versus energy. **c, f** Averaged local anisotropy $A(\mathbf{r}, E)$ around the defects at sublattice $a$ (**c**) and sublattice $b$ (**f**) in region marked as black square in **a** and **d** and as black dashed lines in **b** and **e** versus energy. The energy maxima of the anisotropy are indistinguishable from $E \sim 0$.

versus bias. The anisotropy is very weak (light blue and light red) at the energies outside the superconducting gap, while, inside the superconducting gap, the anisotropy rapidly increases (dark blue and dark red) and its maxima are indistinguishable from $E \sim 0$. Moreover, the curves of $A(\mathbf{r}, E)$ at the second atom site away from the defect center (region marked by black squares in Fig. 4a, d) also exhibits this property (Fig. 4c, f). For comparison, we plot the theoretical curve of $A(\mathbf{r}, E)$ along the same high-symmetry directions at each energy in Fig. S4. The theory curve features the same tendency as the experimental curve

that $A(\mathbf{r}, E)$ is significantly enhanced inside the superconducting gap and the maximum of $A(\mathbf{r}, E)$ is indistinguishable from $E \sim 0$.

Finally, we use a multi-atom (MA) averaging technique resolved by sublattice to establish the repeatability of these phenomena for all equivalent impurity atoms. Figure 5a, b and e, f indicate the scattering centers at sublattice $a$ (Fig. 5a, b) and sublattice $b$ (Fig. 5e, f) marked by red circles that are involved in the MA analysis. The MA technique averages the mapping data over several defects located at the same sublattice[26].

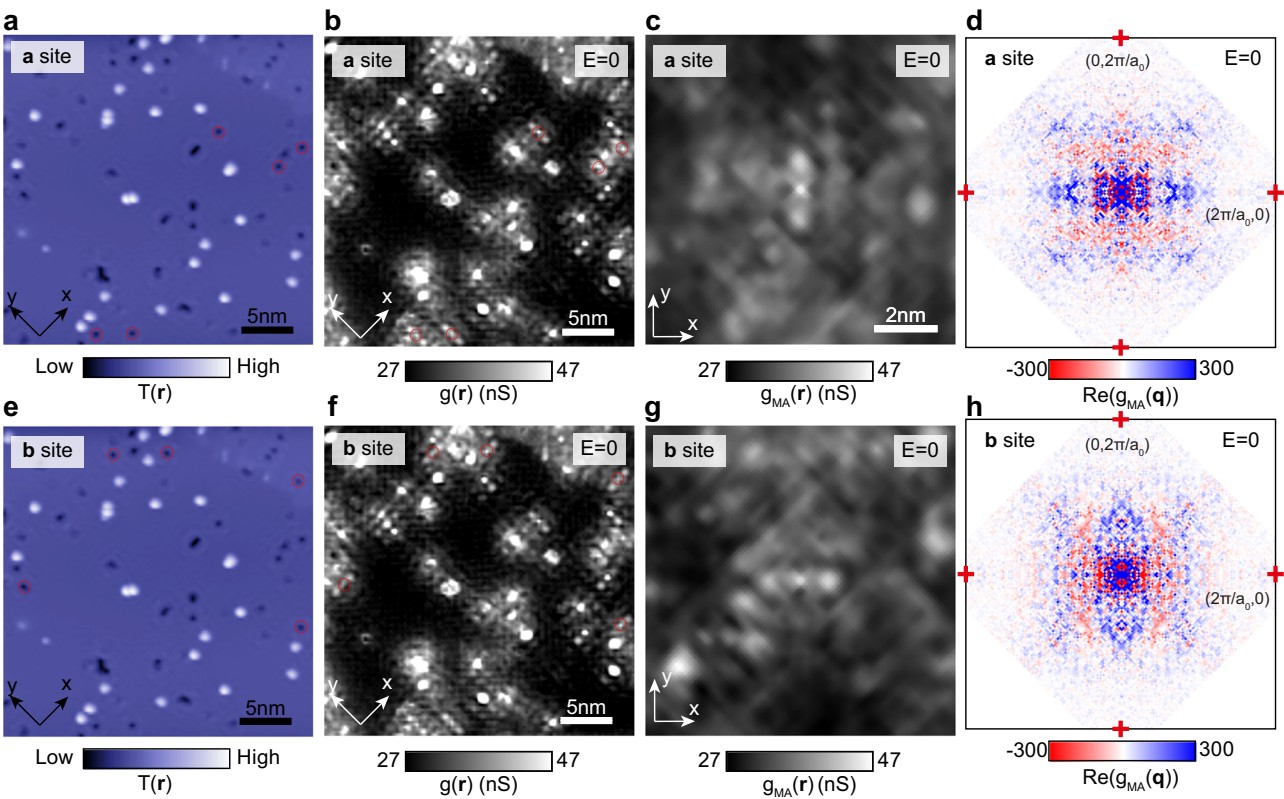

**Fig. 5 | Multi-atom QPI analysis sampled by sublattice. a, e** Topography of the surface of CeCoIn$_5$. (setpoint: V = -10 meV,I = 800 pA). **b, f** Differential conductance map $g(\mathbf{q},E)$ of the surface of CeCoIn$_5$. The scattering centers at sublattice $a$ (**a**, **b**) and sublattice $b$ (**e**, **f**) are marked by red circles, which are involved in the multi-atom analysis. **c,g,** Simultaneous MA-averaged differential conductance map $g_{MA}(\mathbf{r},E)$ at $E \sim 0$ around the defects at sublattice $a$ (**c**) and sublattice $b$ (**g**). **d, h** Real components of Fourier transformed MA-averaged differential conductance map $Re(g_{MA}(\mathbf{q},E))$ at $E \sim 0$ around the defects at sublattice $a$ (**d**) and sublattice $b$ (**h**).

Since multiple sites are averaged, the random local distortion and noise are suppressed, and the common features surrounding the defects are enhanced (SI Section 3). Figure 5c, g present the MA-averaged topography and differential conductance map $g_{MA}(\mathbf{r},E)$ at $E = 0$ for impurity atoms on sublattice $a$ and $b$, respectively. The similar features seen in MA-averaged differential conductance map $g_{MA}(\mathbf{r},E)$ (Fig. 5c, g) and single-defect differential conductance map $g(\mathbf{r},E)$ (Fig. 3c) reveals that the scattering interferences from the two sublattices are highly distinct and repeatably rotated by 90-degrees relative to each other. One advantage of the MA process is that, since the random features in the mapping image are suppressed, the averaged image can be regarded as a single defect that resides in a defect-free large FOV, even though the actual sample is defective. This advantage allows us to Fourier transform the interference signal surrounding the single defect with high resolution in $\mathbf{q}$-space. We set the real-space origin ($\mathbf{r} = \mathbf{0}$) at the defect site and focus on the real part of Fourier transformed map $g_{MA}(\mathbf{q},E)$ (Fig. 5d, h), as the defects are symmetric under the inversion operation and the real part of Fourier terms represent centrosymmetric cosine waves in $\mathbf{r}$-space. Again, $Re(g_{MA}(\mathbf{q},E))$ of the defects at different sublattice $a/b$ is also related to each other by a 90-degree rotation. Remarkably, several features of $Re(g_{MA}(\mathbf{q},E))$, for example the distribution of the positive (blue) and negative (red) values, are reproduced by our theory (Fig. S7). Note that our theoretical model is based on a simple band dispersion, as described in SI Section 1.

## Discussion

In this work we have explored the QPI signatures of ($\pi,\pi$) orbital order in CeCoIn$_5$. The subtlety of such orders is in their preservation of crystal lattice symmetries, which makes them undetectable by traditional scattering techniques[8,9]. On the other hand, pioneering STM

visualization studies of anisotropic electron density due to orbital order has been reported[10,27]. Such experiments must be carried out under extreme tunneling conditions, for example at currents >10 nA that, according to the Tersoff-Hamann theory[28], requiring a miniscule tip-sample distance. Such tip-surface distances usually challenge the stability of the STM junction and, moreover, the tip-sample interaction may then become so intense as to alter the sample properties. By contrast, taking CeCoIn$_5$ ($\pi,\pi$) orbital order as an example, we have explored the possibility of using conventional junction QPI to detect the local symmetry breaking orbital order. From theory, it was predicted that, even with an isotropic impurity, the underlying orbital order should reveal itself as a sublattice-selective anisotropy in the surrounding QPI pattern, due to the different effective coupling of the impurity to the two orbitals. This is because although the impurity is described as a simple point-like potential with no spatial or orbital structure, the scattering T-matrix reflects the orbital order. This suggests strongly that the specific type of impurity is irrelevant to the overall conclusions. While the anisotropy of the scattering interferences is found to be essentially indiscernible in the normal electron state outside the superconducting gap, it is significantly enhanced at energies within the superconducting gap. This finding suggests an interesting effect where the energy scale of QPI experiments used in detection of hidden orbital order is governed by the superconducting gap energy, despite the energy scale of the underlying orbital being much larger. To investigate the prediction experimentally, we performed STM measurement on CeCoIn$_5$, which yields remarkable agreement with the theory. Given our minimal model approach, where only $d_{xz/yz}$-orbitals are considered alongside the anti-ferro-orbital order, superconductivity and a point-like impurity, the agreement with the experimental data is striking and suggests that the methods

may be applicable to a range of superconducting materials exhibiting hidden order[29].

## Methods

### Experiments

Single crystals of $CeCoIn_5$ were synthesized from an In flux by combining stoichiometric amounts of Ce and Co with excess In an alumina crucible and encapsulating the crucible in an evacuated quartz ampoule(details in ref. 18. Its superconductivity and electronic structure were studied in the previous work with $T_c = 2.3 K$ and $\Delta = 600 meV$[19]. The samples were cleaved in ultra-high vacuum at 10 K before inserted into STM. All data are measured by etched tungsten tips with an energy-independent density of states. A standard lock-in amplifier was used for measuring scanning tunneling spectra. See Supplementary Information for additional details on data treatment and extraction.

### Theory

The two-dimensional square-lattice including staggered orbital order and superconductivity has been modeled by the Bogoliubov-de Gennes Hamiltonian in Eq. (1). Simulations of the sublattice-selective Bogoliubov quasiparticle interference have been performed using a T-matrix approach, where a Fourier transformation of (1) allows for a computation of the real-space local density of states (LDOS) in the presence of an impurity using $N(\boldsymbol{R}, \gamma, E) = -\frac{1}{\pi} Im \left( G^R(\boldsymbol{0}, E) + G^R(\boldsymbol{R}, E) T(\boldsymbol{0}, E) G^R(-\boldsymbol{R}, E) \right)_{\gamma\gamma}$.

The impurity was assumed to be non-magnetic with a trivial spatial structure (i.e., point-like). For comparison to experiment all computed $N(\boldsymbol{r}, E)$ are weighed by atomic-like $d_{xz}/d_{yz}$ orbitals and an energy convolution was performed to model the finite experimental energy resolution. Finally, the quasiparticle interference anisotropy was obtained as $A(\boldsymbol{r}, E) = (N(\boldsymbol{r}, E) - N^{\circlearrowleft 90}(\boldsymbol{r}, E))/(N(\boldsymbol{r}, E) + N^{\circlearrowleft 90}(\boldsymbol{r}, E))$. The LDOS anisotropy is strongly enhanced within the superconducting gap as evident from Fig. S4. The full model, all input parameters and further details of the calculations can be found in Supplementary Information Sections 1 and 2.

## Data availability

All data are available in the main text on *Zenodo*[30]. Additional information is available from the corresponding author upon reasonable request.

## Code availability

The simulation code are provided on *Zenodo*[30]. The data analysis codes used in this study are available from the corresponding author upon reasonable request.

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

## Acknowledgements

J.C.S.D. acknowledges support from the Moore Foundation's EPiQS Initiative through Grant GBMF9457, from the European Research

Council (ERC) under Award DLV-788932 and from Science Foundation of Ireland under Award SFI 17/RP/5445. W.C. and J.C.S.D. acknowledge support from the Royal Society under Award R64897. C.N.B., A.K., and B.M.A. acknowledge support by the Danish National Committee for Research Infrastructure (NUFI) through the ESS-Lighthouse Q-MAT. P.J.H acknowledges support from the Department of Energy under Grant No. DE-FG02-05ER46236. C.P. acknowledges support from the U.S. Department of Energy, Office of Basic Energy Science, Division of Materials Science and Engineering, under Contract No. DE-SC0012704 (materials synthesis).

## Author contributions

W.C. and J.C.S.D. conceived the project. J.C.S.D., A.K., and B.M.A. supervised the research. C.P. synthesized and characterized the samples; F.M. and M.P.A. carried out STM measurements. Theoretical analyses were carried out by C.N.B, P.J.H., B.M.A., and A.K. Experimental analysis was carried out by W.C. J.C.S.D. and B.M.A. wrote the paper with key contributions from A.K., C.N.B., P.J.H., and W.C. The manuscript reflects the contributions and ideas of all authors.

## Competing interests

The authors declare no competing interests.
