## [Peer Review File · Nature Communications]

REVIEWER COMMENTS

Reviewer #1 (Remarks to the Author):

In this manuscript W. Chen et al., reported an STM study on CeCoIn₅. They observed that the impurities at two different surface sublattices display anisotropic QPI patterns along orthogonal orientations. This anisotropy is only obvious when the energy is smaller than superconducting gap, but disappears at large energies. Through their model calculation, they found such anisotropy can be well explained by the interplay of orbital order and d-wave superconductivity. Thus this study could demonstrate a new method for studying the "hidden" order which may not be easily detected in other techniques. I found the result is interesting and the logic is clear. I can potentially suggest publication when the issues below are addressed:

1. An important issue is that the surface termination layer is not specified. As shown in previous STM studies (e.g., ref. 10), the cleaved surface of CeCoIn₅ have different terminations such as Co plane and CeIn plane, and they can have the same lattice constant and similar STM image. However, the orbital order is expected to exist in the Co plane. If the surface is CeIn plane, whether it is still detectable? I think it is necessary to specify the two sublattices in Fig. 3(a) that which atomic plane (Co/In/Ce) they are belong to, since different atomic sites obviously have different orbital features.

2. In Fig. 3(c) and Fig. 5(b,f), one can see other defects which show fourfold symmetric QPI pattern. Are they located at different atomic site? It would be helpful to analyze defects on different locations, to see whether they are consistent with the current interpretation (at least in symmetry).

Reviewer #2 (Remarks to the Author):

The authors investigate the quasiparticle interference (QPI) patterns due to an isotropic impurity in the normal and superconducting state of the heavy-fermion compound CeCoIn₅ using STM. Their findings provide QPI patterns with broken four-fold symmetry. Using a minimal model and adopting an approach which appears though not to be self-consistent one, they demonstrate that such patterns may originate from a staggered-orbital order. The quality of this study, which combines both experimental as well as theoretical investigation, is high. The results obtained here are potentially significant in the research field of unconventional superconductors. The materials exhibiting unconventional superconductivity are known to exhibit a variety of ordered states including orbital order. However, there are several concerns, which include the choice of Hamiltonian as well as the extent of advancement made through the current work in comparison to similar works carried out earlier [Singh et al. Sci. Adv. 2015;1:e1500206]. These concerns are mentioned below.

(i) There should be a proper justification for the choice of the model Hamiltonian. The authors consider a model Hamiltonian based on the d_{xz} and d_{yz} orbitals in order to describe the orbital order as well as d-wave superconductivity. However, the staggered orbital order involves $3d_{xz}$ and $3d_{yz}$ orbitals of the Co atoms (Ref [10]) present in the surface layer as indicated by the first-principle calculation. On the other hand, the superconductivity observed in CeCoIn₅ originates through the hybridized bands originating from 4f orbitals of the Ce atom and 5p orbitals of the In atom.

(ii) The detection of orbital order with the help of QPI obtained in the STM measurement is not

new. For instance, the same has been demonstrated in the superconducting state of iron chalcogenides and in the nematic state beyond the superconducting transition temperature [Singh et al. Sci. Adv. 2015;1:e1500206]. The major difference from the previous work is that here QPI is used to detect staggered orbital order in the d-wave superconducting state of CeCoI₅ instead of a ferro-orbital order in the sign-changing s-wave superconducting state of iron chalcogenides. It may be noted that in both the cases, C₄ symmetry in the QPI patterns is broken. Thus, the authors should clarify about the advancement made through current work over the previous work.

(iii) The authors don't describe the contents of the surface layer. Does the surface layer contain Co atoms or is it a simply CeIn layer?

(iv) In order to separate the energy scales of the orbital and superconducting orders, the energy scale of the orbital order is considered well above the superconducting gap, i.e. $\Delta_{oo} \gg \Delta_d$. What may be the justification of such an assumption?

(v) Are the energy bands obtained within the tight-binding model considered in the current work consistent with those in Fig. 1(b)?

Reviewer #1 (Remarks to the Author):

In this manuscript W. Chen et al., reported an STM study on CeCoIn₅. They observed that the impurities at two different surface sublattices display anisotropic QPI patterns along orthogonal orientations. This anisotropy is only obvious when the energy is smaller than superconducting gap, but disappears at large energies. Through their model calculation, they found such anisotropy can be well explained by the interplay of orbital order and d-wave superconductivity. Thus this study could demonstrate a new method for studying the “hidden” order which may not be easily detected in other techniques. I found the result is interesting and the logic is clear. I can potentially suggest publication when the issues below are addressed:

1. An important issue is that the surface termination layer is not specified. As shown in previous STM studies (e.g., ref. 10), the cleaved surface of CeCoIn₅ have different terminations such as Co plane and CeIn plane, and they can have the same lattice constant and similar STM image. However, the orbital order is expected to exist in the Co plane. If the surface is CeIn plane, whether it is still detectable? I think it is necessary to specify the two sublattices in Fig. 3(a) that which atomic plane (Co/In/Ce) they are belong to, since different atomic sites obviously have different orbital features.

We thank the reviewer for flagging this valid point. The cleaved surface we measured in the experiment is indeed Co terminated surface, where the orbital order occurs. This is shown from:

First, figure S9 shows the measured scanning tunneling spectrum on our sample surface. The spectrum presents a dip at ~5meV corresponding to the hybridized gap of local f electrons and itinerant c electrons. This spectrum is indistinguishable from the spectrum measured on Co surface in Ref.[10], except with improved energy resolution. Moreover, the spectrum of CeIn surface in Ref.[10] displays that a density of states at -30meV greater than that at 30meV, again not what we observe on Co surface.

Second, since the orbital order breaks the equivalence of Co sites in sublattice a and b, two degenerate states should appear on the surface. At the interface of these two degenerate states, domain boundaries should form. Ref. [10] reports that the domain boundaries only appear on Co surface, implying that the orbital order occurs only on Co surface. We also observe many domain boundaries on our measured surface (fig. S8), further evidencing that our cleaved surface is Co-terminated surface.

Text change:

Add SI Section4: Identity of Termination Surface and figure S9

Paragraph 7, line2, add “The Co terminated surface is identified from both its tunneling spectra and domain boundaries(SI section4). ”

2. In Fig. 3(c) and Fig. 5(b,f), one can see other defects which show fourfold symmetric QPI pattern. Are they located at different atomic site? It would be helpful to analyze defects on different locations, to see whether they are consistent with the current interpretation (at least in symmetry).

Fig. 3 and Fig. 5(b,f) indeed show many other types of defects. We separate them into three kinds. Multi-atom defects, absorbates and weak defects. We can find all these three defects in Fig. 3(a), marked by red, purple and green circles, respectively (figure below). They are all located at different atomic sites, but, for the following reasons, they are not suitable for the analysis in this work.

- (a) Atomically resolved topography image around two sublattices. (setpoint: $V = -10$ meV, $I = 800$ pA)
- (b) Simultaneous measured differential conductance map $g(r, E)$ at $E = -0.94$ meV in the FOV of image (a).
- (c) Simultaneous measured differential conductance map $g(r, E)$ at $E = 0$ in the FOV of image (a). The BQPI patterns on the two sublattices are clearly distinct and appear to be rotated by 90-degrees relative to each other.

Multi-atom defects consist of several nearby vacancies on the surface. The arrangement of these vacancies and the center of them are nearly random. To prevent the effect of asymmetry of STM tip, we need to find several of the same type of defects but located at different sublattice sites, as we report throughout the main text. The randomness of multi-atom defects will prevent this key step.

Absorbates generate an atomic protrusion on the surface, either from intercalated atoms or gas in the vacuum. Currently, we do not know what forms these absorbates take, but, from STM topography, they exhibit their own anisotropy. This additional anisotropic disturbance should have significant impact on the surrounding QPI and violates our assumption of isotropic scatterers and are therefore not suitable for analysis.

Weak defects show weak signal in both STM topography and tunnelling spectra map. Their QPI signal is too weak to be detected and is usually covered by the stronger QPI signal scattered from other nearby defects. Thus, this kind of defect is also not suitable for analysis.

In the main text, we choose the single on-site defect in CeCoIn_5 for analysis. It has two advantages.

First, this kind of defect is ubiquitous not only in CeCoIn_5 but also in other samples, no matter whether it is a vacancy or substitution. This suggests our method is universal and suitable to be applied to other samples.

Second, the on-site defect is at the high symmetry point of the crystal lattice. This makes the unusual symmetry breaking very easy to detect.

For these reasons, we only focus on these single atom defects, which do break the symmetry, as theory predicts.

Text change:

Paragraph 4, Line 1, add " We choose an on-site impurity as the scattering center, because this kind of impurity widely exists in the crystals and is located at a high symmetry point required to detect the local symmetry breaking caused by the orbital order."

Reviewer #2 (Remarks to the Author):

The authors investigate the quasiparticle interference (QPI) patterns due to an isotropic impurity in the normal and superconducting state of the heavy-fermion compound CeCoIn₅ using STM. Their findings provide QPI patterns with broken four-fold symmetry. Using a minimal model and adopting an approach which appears though not to be self-consistent one, they demonstrate that such patterns may originate from a staggered-orbital order. The quality of this study, which combines both experimental as well as theoretical investigation, is high. The results obtained here are potentially significant in the research field of unconventional superconductors. The materials exhibiting unconventional superconductivity are known to exhibit a variety of ordered states including orbital order. However, there are several concerns, which include the choice of Hamiltonian as well as the extent of advancement made through the current work in comparison to similar works carried out earlier [Singh et al. Sci. Adv. 2015;1:e1500206]. These concerns are mentioned below.

(i) There should be a proper justification for the choice of the model Hamiltonian. The authors consider a model Hamiltonian based on the d_{xz} and d_{yz} orbitals in order to describe the orbital order as well as d-wave superconductivity. However, the staggered orbital order involves $3d_{xz}$ and $3d_{yz}$ orbitals of the Co atoms (Ref [10]) present in the surface layer as indicated by the first-principle calculation. On the other hand, the superconductivity observed in CeCoIn₅ originates through the hybridized bands originating from 4f orbitals of the Ce atom and 5p orbitals of the In atom.

We agree with the referee that the model Hamiltonian does not include the microscopic origin of the superconductivity, nor do we make an attempt to include states from Ce or In atoms.

However, as pointed out by the referee, ab-initio calculations show that the $3d_{xz}$ and $3d_{yz}$ orbitals have significant contribution at the Fermi level. This is consistent with findings in the literature, see for example [Andrzej Ptok et al 2017 New J. Phys. 19 063039]. For convenience, we show the corresponding band structure and density of states of such a calculation.

(a) Band structure at low energies from a fully relativistic DFT calculation for CeCoIn₅ showing many entangled bands. (b) corresponding partial density of states showing a large contribution from the flat bands of Ce character, but significant contribution of Co states as well which are separated out in panel (c) where the Co-3d partial density of states is sizable at the Fermi level.

With these results in hand, let us explain (1) why we are not aiming to construct a more complete model Hamiltonian and (2) why this is not needed for our conclusions.

First, as found in [Andrzej Ptok et al 2017 New J. Phys. 19 063039], the full tight-binding model of CeCoIn₅ should include at least 25 orbitals. Combined with the staggered orbital order this would require an extension from four to 50 orbitals included in the model Hamiltonian. Such an extension is

beyond the scope of the current work and would require a very detailed understanding of the superconducting order parameter in this setting as well.

Second, as argued above, there is evidence for significant density of states from the 3d-states at the Fermi level and these states are hybridized with the states from Ce and In since a projection to 25 orbitals is needed for downfolding. So, it is expected that the superconducting order parameter is also present in the Co-d orbital components, justifying our assumption of a minimal model with superconductivity carried by the d-states. Extending the model to include the Ce-4f orbitals and In-5p orbitals could indeed yield quantitative differences but qualitatively we would still expect a similar symmetry breaking close to the impurities and the enhancement of the symmetry breaking within the gapped energy range from superconductivity. This is simply because the orbital order appears in the Co-d orbitals which, as explained above, also inherit some pairing correlations

Text change:

SI section 1, Final Paragraph, add

“In this work, we only consider the simplest model Hamiltonian including staggered orbital order and it is not identical to the real Fermi surface of CeCoIn₅. We do not discuss a more complete model including both Ce and In atoms and the superconductivity originating from Ce atoms, since such issues are both beyond the scope of our current work and not relevant to its conclusions. Nevertheless, as shown in Fig. S7, the overall pattern of the real part of BQPI is still present in a good agreement between the calculation and the experiment except some inconsistencies in the exact period of the Friedel oscillations. This implies that our model indeed captures the key ingredients of symmetry-breaking QPI induced by the orbital order.”

We involve the change for both concern (i) and (v) in the same paragraph, because both concerns are correlated to the Hamiltonian $\mathcal{H}_0(\mathbf{k})$ chosen in the theory model.

(ii) The detection of orbital order with the help of QPI obtained in the STM measurement is not new. For instance, the same has been demonstrated in the superconducting state of iron chalcogenides and in the nematic state beyond the superconducting transition temperature [Singh et al. Sci. Adv. 2015;1:e1500206]. The major difference from the previous work is that here QPI is used to detect staggered orbital order in the d-wave superconducting state of CeCoIn₅ instead of a ferro-orbital order in the sign-changing s-wave superconducting state of iron chalcogenides. It may be noted that in both the cases, C₄ symmetry in the QPI patterns is broken. Thus, the authors should clarify about the advancement made through current work over the previous work.

The work mentioned by the referee indeed identified nematic order in the FeSeTe system (also by anisotropic QPI pattern). However, there are some fundamental differences to the interpretation of the present data and the mechanism:

First, the FeSeTe has global nematicity, i.e., only one domain visible while in our interpretation of the CeCoIn₅ surface, there is no C₄ symmetry broken globally. Instead, staggered nematicity rotates with the sublattice. In other words, a similar QPI analysis using the density oscillations induced by all impurities will lead to an overall C₄ symmetric signal (since half of the impurities produce rotated QPI). This is the reason why previous global QPI analyses don't notice the existence of the orbital order on the surface of CeCoIn₅. From a technical perspective, it is much easier to detect “global nematicity” compared to a staggered orbital order as discussed in our work.

Second, the cited work discusses the interplay of the observed nematicity with superconductivity. However, nematicity is only observed at energies beyond the coherence peak (which in FeSeTe is roughly at 2meV, see for example Fig. 3 in [Singh et al. Sci. Adv. 2015;1:e1500206]). This is distinct from

the present picture where anisotropy is almost invisible at energies outside the superconducting gap, while at energies smaller than the coherence peaks, the gapping out of some states enhances the nematic signal, thus superconductivity assists in detecting the nematic order, a mechanism to observe the orbital ordering not discussed thus far.

Text change:

SI section 2, Final Paragraph, add

“We point out ref.[Singh et al. Sci. Adv. 2015;1:e1500206] observe the similar symmetry-breaking QPI caused by the nematicity in FeSeTe system. Their observation is distinct from our result in two aspects. First, the global QPI analysis they perform discovers beautifully the order that breaks overall crystal lattice symmetry, but should not yield anti-ferro-orbital order which keeps the global C_4 symmetry as in CeCoIn₅. Second, the nematicity they discover is only observed in the non-superconductive state, high energy beyond the coherence peak. This is also distinct from our present picture that the anisotropy from orbital order is significantly enhanced within the superconductive quasiparticles below the superconducting gap.”

(iii) The authors don't describe the contents of the surface layer. Does the surface layer contain Co atoms or is it a simply CeIn layer?

We appreciate the reviewer raising this point, as Reviewer#1 did. And our reply is the same. The cleaved surface we measured in the experiment is indeed Co terminated surface, where the orbital order occurs. We determine this surface by two key pieces of evidence

First, figure S8 show the measured scanning tunneling spectrum on our sample surface. The spectrum presents a dip at ~ 5 meV corresponding to the hybridized gap of local f electrons and itinerant c electrons. This spectrum is exactly the same as the spectrum measured on Co surface in Ref. [10], except that our energy resolution is better. However, the spectrum of CeIn surface in Ref. [10] displays that the density of states at -30meV are more than that at 30meV, different from what we observed on Co surface.

Second, since the orbital order breaks the equivalence of Co sites in sublattice a and b, two degenerate states should appear on the surface. At the interface of these two degenerate states, domain boundaries should form. Ref. [10] reports that the domain boundaries only appear on Co surface, implying that the orbital order occurs only on Co surface. We also observe many domain boundaries on our measured surface(fig. S8), indicating our cleaved surface is Co-terminated surface.

Text change:

Add SI section 4: Identification of Termination Surface and figure S9

Paragraph 7, line 2, add “The Co terminated surface is determined by both tunneling spectra and domain boundaries(SI section4). ”

(iv) In order to separate the energy scales of the orbital and superconducting orders, the energy scale of the orbital order is considered well above the superconducting gap, i.e. $\Delta_{oo} \gg \Delta_d$. What may be the justification of such an assumption?

As the referee correctly points out, one of the incentives for choosing $\Delta_{oo} \gg \Delta_d$ is indeed to separate ordering energy scales enabling a clear identification of the effect of superconductivity. From an experimental perspective the choice would be justified by data supporting a higher ordering

temperature of the orbital order i.e., $T_{oo} \gg T_d$. Unfortunately, to the best of the authors' knowledge, a study identifying T_{oo} has not been carried out. However, according to Ref. [10], it is claimed that a signature of orbital order is identical over the temperature range 500mK \sim 6 K. Comparing to $T_d \sim 2.3$ K, this makes $T_{oo} > T_d$ a very reasonable assumption. Given this experimental evidence $\Delta_{oo} \sim 3\Delta_d$ combined with the advantage of separating ordering energy scales, we find it justified to let $\Delta_{oo} \gg \Delta_d$.

Finally, we note that the qualitative behavior of all results is identical if $\Delta_{oo} \sim \Delta_d$: Enhancement of the symmetry breaking within the gapped energy range from superconductivity. Below we show the local anisotropy as a function of energy along high symmetry directions (1,0) and (0,1) obtained for $\Delta_{oo} = 0.1|t_1| \sim \Delta_d$. While the maximal anisotropy is decreased by approximately a factor of 2, the results are qualitatively identical to the results shown in Fig. S4 for $\Delta_{oo} = 0.25|t_1|$

Local anisotropy as a function of energy along high symmetry directions (1,0) and (0,1) with $\Delta_{oo} = 0.1|t_1|$. Local anisotropy $A(r, E)$ at two sites away from the impurity along (1,0) (red curve) and (0,1) (black curve) with the impurity positioned at sublattice a (a) and sublattice b (b). Green (blue) curve is the local anisotropy $A(r, E)$ of the model in the normal state along (1,0) ((0,1)) obtained by setting $\{\Delta_1, \Delta_2\} = \{0.0, 0.0\}$. Black dashed lines indicate the energy of superconducting gap Δ .

Text change:

SI section 1, Paragraph 2, Line 16 add "In this model, $\Delta_{oo} \sim 3\Delta_d$, is estimated from the experimental fact that orbital order on the surface of CeCoIn₅ exists even at 6 K while the superconducting temperature of CeCoIn₅ is 2.3K. We also calculate the anisotropy for $\Delta_{oo} = 0.1|t_1| \sim \Delta_d$ (fig. S8). The results are qualitatively identical to the results shown in Fig. S4 for $\Delta_{oo} = 0.25|t_1|$."

Add figure S8.

(v) Are the energy bands obtained within the tight-binding model considered in the current work consistent with those in Fig. 1(b)?

The Fermi surface shown in Fig. 1(b) is adopted from Ref. [19] where the Fermi surface and gap structure of CeCoIn₅ are experimentally determined from the QPI signatures. Given the theoretical choice of considering the simplest model Hamiltonian including staggered orbital order and superconductivity in the Co-d_{xz}/d_{yz} orbitals, the tight-binding model considered here is not identical to the Fermi surface shown in Fig. 1(b).

However, since the main result of the current work is the anisotropy obtained from real-space conductance maps, these results do not depend qualitatively on either the Fermi surface shape nor the magnitude of the Fermi surface wave vector. The details of the Fermi surface will have a strong impact on the QPI patterns and indeed, as shown in Fig. S7, there are inconsistencies between data and computations in the exact period of the Friedel oscillations. Nevertheless, a good agreement of the overall pattern is still present. This implies that our model indeed captures the key ingredients of symmetry-breaking QPI induced by the orbital order.

Additionally, it should be mentioned that we also studied the model Hamiltonian analytically in the Born limit to investigate whether the anisotropy arose from well-known effects e.g., an enhanced joint density of states (JDOS) at certain q-vectors. It was found that the diagonal terms (i.e., the JDOS) of the QPI signal cancel in the expression for the anisotropy, again highlighting the insignificance of the Fermi surface details.

Text change:

Paragraph 3, Line 14, add “Here for generality we consider the simplest model Hamiltonian ($\mathcal{H}_0(\mathbf{k}), \mathcal{H}_{oo}(\mathbf{k})$) rather than specific Hamiltonian of CeCoIn₅”

SI section 1, Final Paragraph, add

“In this work, we only consider the simplest model Hamiltonian including staggered orbital order and it is not identical to the real Fermi surface of CeCoIn₅. We don’t discuss a more complete model including both Ce and In atoms and the superconductivity originating from Ce atoms, since such issues are both beyond the scope of our current work and not relevant to its conclusions. Nevertheless, as shown in Fig. S7, the overall pattern of the real part of BQPI is still present in a good agreement between the calculation and the experiment except some inconsistencies in the exact period of the Friedel oscillations. This implies that our model indeed captures the key ingredients of symmetry-breaking QPI induced by the orbital order.”

We involve the change for both concern (i) and (v) in the same paragraph, because both concerns are correlated to the Hamiltonian $\mathcal{H}_0(\mathbf{k})$ chosen in the theory model.

Additional text change:

Change figure 1. Increase the δ band in fig. 1b, as described in the main text.

REVIEWERS' COMMENTS

Reviewer #1 (Remarks to the Author):

I have read the author's reply and the revised manuscript. The authors provided additional data of tunneling spectrum and scanning images (Fig. S9) to show that the cleaved surface is Co-terminated. The conclusion is mainly based on observation of domain structures on the surface (the tunneling spectrum does not show distinct features between different surfaces, as mentioned in ref. 10). I consider the conclusion acceptable, however I would appreciate if the authors can provide more details on the domain wall structure, which will further prove the domains are really induced by orbital orders on the two sublattice. The authors reply to the other comments is satisfactory.

Reviewer #2 (Remarks to the Author):

I believe that the authors have addressed adequately the issues raised by the referees. Therefore, the manuscript in the current form may be published in Nature Communications.

REVIEWERS' COMMENTS

Reviewer #1 (Remarks to the Author):

I have read the author's reply and the revised manuscript. The authors provided additional data of tunneling spectrum and scanning images (Fig. S9) to show that the cleaved surface is Co-terminated. The conclusion is mainly based on observation of domain structures on the surface (the tunneling spectrum does not show distinct features between different surfaces, as mentioned in ref. 10). I consider the conclusion acceptable, however I would appreciate if the authors can provide more details on the domain wall structure, which will further prove the domains are really induced by orbital orders on the two sublattice. The authors reply to the other comments is satisfactory.

We thank the reviewer for the suggestion. Ref. 10 already provides the atomic image of the domain wall. The figure below in our data can prove that the domains and domain boundary are indeed induced by orbital orders on the two sublattice.

a. CeCoIn₅ topography with two domains near a domain boundary. **b.** The same topography in a with adjusted colormap limit to show the atom sites. **c.** CeCoIn₅ $g(r, E = 0)$ in the same field of view in **a**. **d.** The schematic diagram of the arrangement of atoms with orbital order marked by red dots(sublattice a site) and blue dots(sublattice b site), according to **b,c**. The atoms at the domain boundary are marked by black dots. The hollow circles show the position of the defects, corresponding to the defects marked in **a,b,c** by blue(sublattice a site) or red circles(sublattice b site).

This figure shows two nearby domains with several defects close to the domain boundary. In the $g(\mathbf{r}, E = 0)$ map (Fig. c) in the same field of view in Fig. a, we choose the same type of the defects in the main text fig. 3 and 5, which apparently break the local C_4 symmetry in the superconducting state at $E=0$, two in the left domain (domain 1) and one in the right domain (domain 2). According to their local anisotropy, we can distinguish at which sublattice site these defects are located, and, then, extract the sublattice a/b site order in each domain (red and blue dot in Fig. d). On the other hand, in Fig. b, the arrangement of Co atoms near the domain boundary can be directly visualized after we adjusted the colormap limits. Finally, in Fig. d, we draw a schematic diagram of sublattice a/b orders near the domain boundary, combining both the arrangement of the atoms shown in Fig. b and the sublattice a/b site order in each domain extracted by Fig. c. It clearly shows that the sublattice a/b site order in the two domains are opposite. This confirms that the domain boundary indeed forms at the interface of two degenerated orbital order states.

Text change:

Add figure S10.

SI section 4, Final Paragraph, add

“ Furthermore, Fig. S10 shows two nearby domains with several defects close to the domain boundary. In the $g(\mathbf{r}, E = 0)$ map (Fig. S10c) in the same field of view in Fig. S10a, we choose the same type of the defects in the main text fig. 3 and 5, which apparently break the local C_4 symmetry in the superconducting state at $E=0$, two in the left domain (domain 1) and one in the right domain (domain 2). According to their local anisotropy, we can distinguish at which sublattice site these defects are located, and, then, extract the sublattice a/b site order in each domain (red and blue dot in Fig. S10d). On the other hand, in Fig. S10b, the arrangement of Co atoms near the domain boundary can be directly visualized after we adjusted the colormap limits. In Fig. S10d, we draw a schematic diagram of sublattice a/b orders near the domain boundary, combining both the arrangement of the atoms shown in Fig. S10b and the sublattice a/b site order in each domain extracted by Fig.S10c. It clearly shows that the sublattice a/b site order in the two domains are opposite. This confirms that the domain boundary indeed forms at the interface of two degenerated orbital order states.”

Reviewer #2 (Remarks to the Author):

I believe that the authors have addressed adequately the issues raised by the referees. Therefore, the manuscript in the current form may be published in Nature Communications.

We acknowledge and thank the editor and reviewers for their positive and constructive comments and suggestions on our manuscript.

Additional Text change:

Some changes in the text to meet the format requirement of nature communication.

Update the Zenodo database including both data and code.